# *Allium*-Based Phytobiotic for Laying Hens’ Supplementation: Effects on Productivity, Egg Quality, and Fecal Microbiota

**DOI:** 10.3390/microorganisms10010117

**Published:** 2022-01-06

**Authors:** Edmundo Ruesga-Gutiérrez, José Martín Ruvalcaba-Gómez, Lorena Jacqueline Gómez-Godínez, Zuamí Villagrán, Victor M. Gómez-Rodríguez, Darwin Heredia-Nava, Humberto Ramírez-Vega, Ramón Ignacio Arteaga-Garibay

**Affiliations:** 1Centro Universitario de los Altos, Universidad de Guadalajara, Av. Rafael Casillas Aceves #1200, Tepatitlán de Morelos, Jalisco 47600, Mexico; edmruesga@gmail.com (E.R.-G.); blanca.villagran@academicos.udg.mx (Z.V.); victor.gomez@cualtos.udg.mx (V.M.G.-R.); darwin.heredia@cualtos.udg.mx (D.H.-N.); 2Centro Nacional de Recursos Genéticos, Instituto Nacional de Investigaciones Forestales, Agrícolas y Pecuarias, Boulevard de la Biodiversidad #400, Tepatitlán de Morelos, Jalisco 47600, Mexico; ruvalcaba.josemartin@inifap.gob.mx (J.M.R.-G.); gomez.lorena@inifap.gob.mx (L.J.G.-G.)

**Keywords:** garlic, high-throughput sequencing, PICRUSt, poultry, *Salmonella* Pullorum

## Abstract

The poultry industry is constantly demanding novel strategies to improve the productivity and health status of hens, prioritizing those based on the holistic use of natural resources. This study aimed to assess the effects of an *Allium*-based phytobiotic on productivity, egg quality, and fecal microbiota of laying hens. One hundred and ninety-two 14-week-old Lohmann Lite LSL hens were allocated into an experimental farm, fed with a commercial concentrate with and without the *Allium*-based phytobiotic, and challenged against *Salmonella*. Productivity, egg quality, and fecal microbiota were monitored for 20 weeks. Results showed that the phytobiotic caused an increase on the number of eggs laid (*p* < 0.05) and in the feed conversion rate (*p* < 0.05); meanwhile, egg quality, expressed as egg weight, albumin height, haugh units, egg shell strength, and egg shell thickness remained unchanged (*p* > 0.05), although yolk color was decreased. Fecal microbiota structure was also modified, indicating a modulation of the gut microbiota by increasing the presence of Firmicutes and Bacteroidetes but reducing Proteobacteria and Actinobacteria phyla. Predicted changes in the functional profiles of fecal microbiota suggest alterations in metabolic activities that could be responsible for the improvement and maintenance of productivity and egg quality when the phytobiotic was supplemented; thus, *Allium*-based phytobiotic has a major impact on the performance of laying hens associated with a possible gut microbiota modulation.

## 1. Introduction

Gut microbiota, represented by bacteria, archaea, viruses, and eukaryotes, take part in regulating several functions in both humans and animals [1]. It has been estimated that the chicken gut can be colonized by at least 1000 different microbial strains that are involved in maintaining the intestinal health of hens while sustaining productivity and growth [2,3]. This microbiota takes part in various metabolic pathways related to nutrient digestion and absorption, including the production of short-chain fatty acids (SCFA) [4], which are important in several processes such as pH reduction and intestinal absorption of nutrients, among others [5]. The immune system development also depends on the gut microbiota composition and structure, providing the host with a large variety of immunogens that stimulate the immune response of the intestinal mucosa and maturation of immune organs [6]. In hens, a stable gut microbiota is generally observed from the fourth week of life, and remains that way unless there is a disruption associated with an abrupt change in the diet, infections by pathogens (including *Salmonella* infection), or some other exogenous factor, inducing metabolic disorders that consequently modify the health and productivity status of hens, including the egg-laying status [7,8].

Many plant species and their fractions have been used as phytobiotics [9], the most popular of which being alfalfa [10], bergamot oil [11], peppermint [12], black cumin [13], chili [14], clove [15], oregano [16], cinnamon [17], and garlic [18], among others [19]. Antimicrobial effects and microbiota modulation associated with phytobiotics also could involve cecal metabolic changes in poultry, as has been observed when chestnut tannins were supplemented to broilers, which exhibited an increase in IL-6 and IL-10 expression in ceca between day 2 and 6 of supplementation. Besides, a kinome array revealed that supplementation with 1% of chestnut tannins in broilers also induced significant changes in immune and metabolic pathways at day 6 in comparison with those registered at day 3, mainly related to phosphate-containing metabolic processes, primary metabolic process, protein metabolic processes, and cellular metabolic processes, among others [20].

Specifically, extracts from *Allium* species, including garlic and onion, have been proposed as a diet supplement for its antibacterial [21], antiviral, and immunostimulatory effects [22], alone or in combination with other compounds, and have shown effects on the mRNA expression of toll-like receptors, along with improvements in weight gain, feed efficiency, and lipid profile in broilers [19,23]. Phytobiotic properties of garlic and onion are mainly related to their content of polyphenols, saponins, fructans, organosulfur compounds and fructooligosaccharides [24,25,26]. This study aimed to assess the effects on productivity, egg quality, and microbiota modulation of laying hens supplemented with an *Allium*-based phytobiotic; therefore, we would increase productivity and improve egg quality by modifying the intestinal microbiome of laying hens. 

## 2. Materials and Methods

### 2.1. Phytobiotic Mixture

A commercial *Allium*-based phytobiotic (GOL^TM^, Bavaria Corp. International, Apopka, FL, USA), consisting mainly of organosulfur compounds, was used following the manufacturer’s recommendation (100 g per ton ratio to the diet for laying hen supplementation). 

### 2.2. Hens Housing and Feeding

The study was carried out using 192 14-week-old Lohmann Lite LSL laying hens allocated into an experimental farm in Tepatitlán de Morelos, Jalisco, México (20°55′12.91″ N and 102°39′05.64″ W, 1850 masl) under the approval of a bioethics and biosafety committee (CUALTOS) of the University of Guadalajara (CUA/CEL/131,120). Hens were housed in a population density of 750 cm^2^ per bird in a battery-system cage (two birds per cage) with an average of 25 cm of available feeder per bird. Allocation was isolated and provided with HEPA filtered air and under negative pressure. The room temperature ranged between 18 and 20 °C. Birds were exposed to 16 h of light per day. Basal diet consisted of a commercial feed (Protein: 18%, Calcium: 3.8%, Available Phosphorus: 0.5%, D-Lysine: 0.92%, L-Methionine: 0.55% ME, kcal/kg: 2929). An adaptation period of seven weeks before the start of the experimental procedures was considered. At twenty-one weeks, birds were divided into four groups: Control Group (G1), Group 2 (G2), Group 3 (G3), and Group 4 (G4), with eight subgroups of six birds each. Hens in G1 and G4 were fed only with the commercial feed, meanwhile birds in G2 and G3 were also supplemented with the phytobiotic mixture at a ratio of 100 g of the phytobiotic per ton of commercial feed. The nutritional plan was based on the nutritional requirements suggested in the Lohmann Lite LSL management guide. Feed and water were offered ad libitum. 

### 2.3. Salmonella Infection

Layers corresponding to G3 and G4 were intentionally infected with the strain *Salmonella enterica* subsp. *enterica* ser. Gallinarum biovar Pullorum ERG19, previously isolated and identified. Infection was at 21 weeks old and performed by an intravenous application of 0.5 mL per bird of a suspension prepared from fresh biomass (24 h) of the bacterial strain at a concentration of 6 × 10^8^ UFC ml^−1^ in a sodium chloride solution (0.85% *w*/*v*).

### 2.4. Feed Intake, Productivity and Egg Quality

Feed intake was calculated daily and expressed as the difference between the amount of the offered feed and the leftover feed divided by the number of birds per feeder. Productivity was measured by daily egg production and average egg weight. Efficiency, expressed as feed conversion index, was estimated by dividing the number of kilograms of feed consumed by the total kilograms of egg produced. Finally, at weeks 21, 31, and 41, twelve eggs per treatment were randomly taken to assess egg quality, expressed as egg weight, yolk color and height, Haugh units (protein quality), egg shell strength, and egg shell thickness. Egg quality determinations were obtained using a Digital Egg Tester DET6000 (NABEL Co., Ltd., Kyoto, Japan).

### 2.5. Fecal-Associated Bacterial Communities

#### 2.5.1. Sampling and DNA Extraction

Fresh hen’s manure samples mixture of each treatment were taken at weeks 21, 31, and 41. Metagenomic DNA from 0.2 g of each of the collected samples was obtained using a commercial extraction kit (Fecal DNA extraction Kit, Bio Basic Inc., Markham, ON, Canada). DNA integrity was verified through a 1% agarose gel electrophoresis for 50 min at 80 V. DNA was stored at −20 °C until used in sequencing procedures.

#### 2.5.2. Preparation of Libraries and Sequencing Procedure

The construction of 16S DNA libraries started with the PCR-based amplification of 7 of the 9 hypervariable regions of the 16S rDNA gene (V2, V3, V4, V6–V9), achieved in two independent reactions throughout the use of the 16S metagenomics system following the manufacturer’s instructions (Thermo Fisher Scientific, Waltham, MA, USA) in a SelectCycler device (Select BioProduct, Life Science Research, Waltham, MA, USA). Afterwards, 50 nanograms of the equimolar mixture prepared from the amplification products were used to generate the 16S rDNA libraries with the Ion Plus Fragment Library commercial system and the Ion Xpress barcode adapters (Thermo Fisher Scientific). Libraries were purified with the Agentcourt AMPure XP system according to the manufacturer’s instructions (Beckman Coulter, Brea, CA) and quantified using a highly sensitive DNA commercial system and the Bioanalyzer 2100 (Agilent Technologies, Santa Clara, CA, USA). Concentration was adjusted to 26 pM followed by PCR-amplification of the PCR emulsion using a volume of 25 µL of the equimolar mixture of all samples (One-Touch 2, Thermo Fisher Scientific) and enriched with the OneTouch Enrichment system (Thermo Fisher Scientific). Sequencing was performed using the Ion S5™ system (Thermo Fisher Scientific).

#### 2.5.3. Bioinformatics

Resulting files, from the sequencing with Ion Torrent, were converted from BAM format to FASTQ files to evaluate the quality of the sequences using FastQC. Quality control and processing were performed through QIIME2 (v.2021.4.0) [27]. The sequences were demultiplexed using a Q score of 30; meanwhile, the adapters from the sequencing barcodes were removed with cutadapt [28]. Finally, chimeric and low-quality sequences were removed with DADA2 [29]. Clean sequences were compared against the Greengenes database (gg-13-8-99-515-806-nb-classifier.qza) followed by taxonomic assignment (ASVs). The alpha diversity indexes (observed species, Chao1, Shannon, and Simpson) were calculated from the rarefaction curves. Beta diversity indexes were calculated using the Euclidean distance and UniFrac (weighted and unweighted) were visualized through principal coordinate analysis (PCoA). This analysis was performed within qiime2; the Euclidean distance measures the Euclidean distances between the replica species. We used the single-branch-length fraction through unweighted_unifrac and the neighbor joining method was used for clustering. Finally, a predictive functional analysis was obtained through Phylogenetic Investigation of Communities by Reconstruction of Unobserved States (PICRUSt, v. 2.4.1) [30] analyzed on MetaCyc [31]. The sequences of the 16S rRNA gene derived from this study are available from the NCBI under the Bioproject access id PRJNA792576.

#### 2.5.4. Experimental Design

A completely randomized experimental design was used. Data were analyzed through ANOVA and means comparisons using the Tukey test (*p* < 0.05). The Statistical software SAS 9.0 (SAS Institute Inc., Cary, NC) was used.

The linear representation of the statistical model was:Yij = μ + τi + εij(1)
where: Yij is the dependent variable, μ is the general mean, τj is the treatment effect, and εij is the random error with mean 0 and variance σ2.

## 3. Results

An *Allium*-based phytobiotic mixture was successfully supplemented to *Salmonella* challenged and not-challenged Lohmann Lite LSL laying hens in order to assess the effects on productivity and the profile of fecal-associated bacterial communities. 

### 3.1. Hens’ Productivity

Productivity of phytobiotic supplemented and not-supplemented laying hens is shown in Table 1.

The unsupplemented and unchallenged group (G1) and the phytobiotic supplemented but unchallenged group (G2) exhibited the highest egg production scores. On the other hand, the G4 group (*Salmonella*-challenged, phytobiotic-unsupplemented hens) exhibited an 8.28% lower egg production (*p* < 0.05) in comparison with the G1 group, although productivity remained closer to the control group when *Salmonella*-challenged birds were supplemented with the phytobiotic mixture (G3). Meanwhile, feed intake was higher in the G1 group (*p* < 0.05) in comparison with the rest of the treatments. G2 and G4 groups showed similar feed intake values. However, feed conversion index scores, which represent the efficiency in terms of feed intake and egg production, were statistically equal for G1, G2, and G3; meanwhile, less efficiency was observed in hens allocated in G4, which recorded an average feed conversion index of 2.50. No significant differences (*p* > 0.05) were observed for mortality. 

### 3.2. Egg Quality

Egg quality scores are listed in Table 2. The albumin height was affected in the treatment G2, unlike treatments G3 and G4, which had a similar behavior (*p* < 0.05). The color of the yolk of eggs produced by hens in G2 and G3 groups was affected by the use of the phytobiotic (*p* < 0.05); meanwhile, no significant differences were registered among groups (*p* > 0.05) for Haugh units, resistance to strength, and shell thickness.

### 3.3. Fecal-Associated Bacterial Communities’ Profile

Amplicons were obtained from nine samples corresponding to the different treatments (20-day Control, *Salmonella*, Phytobiotic supplemented + *Salmonella*, Phytobiotic supplemented, 30 and 40-day Control), and they were analyzed as described above. A total of 2,919,529 readings were obtained with an average of 324,392 readings per sample. After filtering, we obtained 2,171,303 readings and 3752 ASVs were identified; these were assigned to five *phyla*, 10 classes, 14 orders, 23 families, and 32 bacterial genera. Diversity indexes, calculated from the rarefaction curves, remained constant in the different treatments, showing a slight change in the control treatment at week 21 (Table 3).

The most abundant *phyla* in most of the samples were Firmicutes with an average relative abundance of 77%, and it was present in all samples, followed by bacteroidetes with 16% and actinobacteria with 5% (Figure 1A). Something remarkable is that in G1, samples corresponding to 21-week-old hens (beginning of the experimental procedures) scored a high percentage of Actinobacteria (25%) that significantly decreased in G2, G3, and G4, but remained in high abundance in G1 (up to 12% of relative abundance). On the other hand, in G2, G3, and G4, Actinobacteria relative abundance was low, ranging from 0.4% to 5%. Moreover, the Bacteroidetes *phylum* in the 21-week-old hens from G1 registered a low relative abundance (2% average), which increased at week 41, reaching an average relative abundance around of 20%. Meanwhile, in G2, G3, and G4, the average relative abundance of the Bacteroidetes *phylum* at week 21 ranged between 20% and 28%, scores that were affected by phytobiotic supplementation more than by the *Salmonella* infection (Figure 1A).

The predominant class in fecal samples was Bacilli. In the 21-week G1 samples, there was also a high abundance of Actinobacteria (average relative abundance of 25%), while this class was found in lower proportions in G2, G3, and G4 samples (0.3 to 12%). All samples, except the 21-week G1 sample, exhibited a significant relative abundance of the Bacteroidia (average among 2% and 28%) and Clostridia classes (2% to 13% of relative abundance). At genus level (Figure 1C), the dominant group, in most of the samples, was *Lactobacillus* (average relative abundance of 64%), followed by *Bacteroides* (average relative abundance of 12%), *Corynebacterium* (average relative abundance of 4%), and *Rumminococcus* (average relative abundance of 3%). The *Corynebacterium* genus was only observed in the 21-w and 41-w G1 samples. Samples corresponding to G1 at 21 w also registered a high relative abundance of *Enterococcus* (relative abundance near to 57%) but poorly represented in the rest of the samples.

The beta-diversity analysis, based in PCoA, allowed us to corroborate differences in the fecal microbiota structure associated with supplemented or non-supplemented hens and time of exposure to the phytobiotic, as well as *Salmonella*-challenged and unchallenged birds (Figure 2).

Regarding the functional profile of bacterial communities, 147 METACYC pathways were predicted, including all samples. At the level of the functional category, pathways related to the degradation of aromatic compounds, carbohydrate biosynthesis, carbohydrate degradation, and amino acid metabolism were observed. The most representative pathways based on relative frequency in G1 at 21-w corresponded to genes involved in syringate, protocatechuate, catechol, nicotinate, vanillin, vanillate, L-arginine, and L-ornithine degradation. A remarkable metabolic change was predicted at week 41, mainly defined by polyamine biosynthesis and pyrimidine ribonucleosides degradation, as we observed in G1 at 41 w. Moreover, in the G2-41w sample, genes involved in degradation pathways of some compounds such as chlorosalicylate and L-rhamnose were identified, as well as genes related to the biosynthesis of archaetidylinositol and sucrose; whilst, in the sample corresponding to G3 at 41 w, nitrobenzoate, L-tryptophan, toluene, catechol, and methylcatechol degradation pathways were mostly registered. Finally, in samples corresponding to G4 at 41 w, genes involved with the aerobactin biosynthesis were predicted (Figure 3).

Finally, correlation analysis showed that the relative abundance of the Firmicutes *phylum*, strongly represent in the fecal microbiota, correlated with the relative frequency of the main productive parameters and egg quality. For example, Firmicutes positively correlated (*p* < 0.05) with egg production and feed conversion index ratio and negatively correlated with yolk color and egg shell thickness. The Bacterioidetes positively correlated (*p* < 0.05) with egg weight and albumin height; on the other hand, this *phylum* negatively correlated with egg production, egg shell strength, and egg shell thickness (Figure 4).

## 4. Discussion

*Allium*, a genus that involves more than 600 different species, is strongly represented by garlic, onions, leeks, and shallots, besides being the most technologically exploited species. Extracts obtained from several *Allium* species are rich in organosulfur compounds and they have been recognized as antimicrobial compounds, besides in other applications, such as for anti-inflammatory purposes [32]. In general, phytobiotics involve the use of herbs, spices, or extracted oils that have been reported to be useful to improve feed intake, stimulate the secretion of endogenous enzymes, and inhibit pathogens proliferation, improving nutrients absorption via increasing the permeability of cell membranes, or increasing the carcass quality and muscle yield in broiler, to name a few [33].

In this study, the supplementation of laying hens with a phytobiotic comprising a blend of onion and garlic extracts enhanced the productivity of laying hens, even when they were exposed to *Salmonella* Pullorum. In this context and according to a meta-analysis performed in 2021 [34], which summarized the effects on the productivity of garlic supplemented laying hens, garlic supplementation could represent improve on hen day egg production (increase of 0.322%) and egg mass (increment of 0.486 g/day/hen). Besides, egg weight was improved by 0.069 g on average as well as egg shell thickness and egg shell weight (increments of 0.259 mm and 0.219 g, respectively) through garlic-based supplementation; meanwhile, feed conversion index remained constant with or without garlic-based supplementation, as well as feed intake and Haugh unit [34]. 

On the other hand, results showed that the use of the *Allium*-based phytobiotic had no effect on the weight of the egg, resistance of the shell, and the thickness of the shell in contrast to the reported in the referred meta-analysis, although coincides with the reported effects on albumin quality, expressed as Haugh units. However, no effects on egg production, shell-less eggs’ rate, or egg weight have been previously reported when red garlic cloves are used in laying hens feeding or the combination of garlic and onion extracts [35,36]. Other effects have been informed by the use of other growth promoters, phytobiotics, organic acids, and their combination [37]; for example, the use of a phytobiotic mixture composed by ginger rhizome, liquorice, ashwagandha roots, black seeds, and leaves of green tea in the supplementation of one-day-old chickens, alone or in combination with organic acids, resulted in an increase in feed intake and body weight in the phytobiotic-supplemented birds, as well as improving the villus height and villus-height/crypt-depth ratio in ileum, jejunum, and duodenum while down regulating the ghrelin gene expression. Results also suggested that *Lactobacillus* spp. abundance positively impacts villus-height/crypt-depth ratio in duodenum.

The effects of a phytobiotic may also be improved through an adequate age of supplementation. In this context, the addition of natural lavender essential oil in the diet of one-day-old broiler chicken had an improvement when supplemented between day 22 and 42 (more than the observed since day 1 to day 42). Effects were reported on weight gain and feed conversion ratio, besides reducing the number of pathogenic microorganisms and increasing the probiotic bacteria in ileum microbiota; meanwhile, no effects were observed on feed and water intake, survival rate, or blood biochemistry. Concluding that a bird’s age is a crucial factor to decide when supplementation begins and, for economic reasons, it could be more profitable when is performed from rearing day 22 to 42 [38]. Within this framework, our results suggested an increase in productive parameters from week 10 of supplementation when 14-w old hens were used; nonetheless, other technologies have been validated in newborn birds to evaluate effects in growth rate and the beginning of the productive stage.

However, phytobiotics-attributable effects seem to be dependent on the type of compound and the animal species, as was observed when oregano and Enviva essential oils were used as phytobiotics in 11-day-old ducks’ supplementation for 40 days. The study did not exhibit significant effects on the final body weight, body gain weight, growth rate, feed intake, feed conversion index, or serum variables (protein, albumin, globulin, cholesterol, triglycerides, alanine aminotransferase, and aspartate aminotransferase) in ducks; although an antimicrobial effect under populations of total coliforms, total aerobes, and lactose-negative *Enterobacteria* was recorded [39]. 

The use of granulated garlic in the feeding of Hisex-Brown hens increased egg production and their mass, obtaining an improvement in feed conversion [40], while the conversion index in ISA-Brown hens supplemented with 500 g/ton of garlic obtained a feed efficiency of 2.05 [41]. In contrast, Ayed [36] did not find significant differences in the addition of garlic cloves in feed in response to productive parameters such as production and egg weight; however, with 500 g/Ton it obtained a yolk color of 7.06. The inclusion of a phytobiotic mixture of garlic and ginger at a dose greater than 7.5 g/kg improved the feed conversion, and at a dose of 12.5 g/kg, increased the Haugh units [42]; meanwhile, the addition of *Allium* sativum in the feed of laying hens did not show improvements in production index and Haugh units. However, there were improvements in daily egg production, egg weight, and shell thickness [34]. On the other hand, the used fraction or the extraction procedure to prepare the phytobiotic plays a role in the observed effects, as reported by the use of an essential oil blend, which includes garlic, that increased the egg production by 3.6% in birds older than 46 weeks [43]. The use of a liquid garlic extract increased the egg weight and egg mass in Hi-Sex hens of 40 weeks, with no changes in productivity, besides that the weight of the albumin and Haugh units were improved [44]. The combination of garlic and thyme, used in layers supplementation, did not mean changes in egg mass, production, and feed conversion index, but the color of the yolk increased [45]. 

Some other feed additives, included ß-carotene, curcumin, allicin, and butyric acid in the breeder feed, reflected improvements in the productive development of chickens from one day to twenty-eight days of age [46]. The inclusion in the feed of 20 g/kg of peppermint leaves in Hyline-Brown hens of 64 weeks improved the productive parameters which were reflected in a lower feed conversion index [12].

Moreover, high-throughput sequencing-based studies have elucidated that healthy chickens’ gut microbiota changes at different points in the digestive tract; nonetheless, the main observed *phyla* in that organ are Firmicutes, Proteobacteria, and Bacteroidetes whose relative abundance changes according to the portion of the tract; for instance, the small intestine is mainly colonized by Firmicutes (83.9%) and Proteobacteria (13.8%). Meanwhile, large intestine *phyla* are mainly represented by Bacteriodetes (55.3%) and Firmicutes (43.0%) [47,48]. Several factors have been recognized as drivers in gut microbiota composition in poultry, including the age of the birds, diet, the use of antibiotics, some stressors, as well as some strategies that can be incorporated for the modulation of this microbiota such as the use of probiotics, prebiotics, postbiotics, phytobiotics, among others [49], which could explain changes in fecal microbiota observed in this study, as an indicator of gut microbiota modulation induced by the phytobiotic supplementation. 

Nevertheless, gut microbiota composition in hens is not only determined by feeding-based strategies, but can also be a consequence of their behavior and allocation conditions, as reported by Van de Eijk for luminal microbiota (combined content of ileum, ceca, and colon) in high-feather-pecking laying hens that differentiate from low-feather peckers by a higher relative abundance of bacteria genera related to *Clostridiales*, but lower incidence of *Lactobacillus* and *Staphylococcus*, although it not yet elucidated if these differences are causal or a consequence of feather pecking [50]. Another commonly observed conditioning factor is the incidence of infectious agents in the flock, so it is important to highlight the protective effect that phytobiotics may provide, such as the observed in *Campylobacter hepaticus*-challenged hens supplemented with Isoquinoline alkaloids [50]. Results of that study showed that after 4 weeks of supplementation with isoquinoline alkaloids, hens exposed to *C. hepaticus* HV10 showed less miliary lesions on the liver surface and lower lesion scores in comparison with exposed but not-supplemented hens with the phytobiotic. Phytobiotic-treated hens also registered an increase in the Firmicutes/Bacteroidetes ratio of cecal microbiota and reduced the IL-8 production, although they did not exhibited changes in intestinal villus height and crypt depth. Besides, supplemented and challenged hens did not record egg mass reduction, as observed in unsuplemented and challenged birds. Aljumaah et al. also reported an increase in productive parameters using a sanguinarine-based phytobiotic on broiler recovery from necrotic enteritis infection. Although sanguinarine, a benzophenanthridine extracted from many plants, showed no differences in the performance, livability, and histological measurements between supplemented and not-supplemented necrotic enteritis challenged broilers; additionally, a significant improvement was observed in necrotic enteritis lesion score of ileum and duodenum in the sanguinarine-supplemented group, as an indicator of a better post necrotic enteritis recovery associated with the phytobiotic that further had an effect of cecal microbiota through the recovery of Firmicutes-Bacteroidetes ratio modified in necrotic enteritidis challenged and not-supplemented broiler, with an increase of cecal acetic acid production [51].

Another desirable effect when phytobiotics are supplemented to poultry is a protective effect against microorganisms causing poultry diseases. A mechanism of action of phytobiotics has been proposed and consists of the inhibition of bacterial communication (Quorum sensing), which may be a feasible alternative to control bacteria resistant to antibiotics [9], including some strains of *Salmonella*. *Salmonella*, the genus causing salmonellosis in poultry, comprises two species (*S. enterica* and *S. bongori*) and six subspecies classified, at the same time, in serovars, being two of the most important *Salmonella* Pullorum and *Salmonella* Gallinarum [52]. Although chickens are considered as chronic carriers of *Salmonella*, causing both direct and indirect losses (including morbidity, mortality, reduction of growth rate, drop in egg laying, and poor egg quality, among others) [53], its control is essential to avoid as much as possible its negative effects on the poultry industry. In that sense, it has been reported that garlic exhibits antimicrobial effects against *Salmonella* when used in concentrations above 30 mg/mL [54]; besides, garlic supplementation has shown an improvement in weight gain and feed conversion in birds [55]. The inclusion of 500 g/Ton of Biotronic^®^ (a commercial blend of organic acids) improved the weight gain of young birds from zero to eight weeks [56], as well as the inclusion of 400 g/Ton of garlic in feed improved the start of production [35]. Other Phytobiotics have improved the feed conversion index in Ross chickens when challenged against *Salmonella* Typhimurium [33]. Active compounds in phytobiotic mixtures improve the conversion rate and egg production [57]. The use of different compounds of natural origin improves the feed conversion in Ross 308 chickens when faced with a challenge from *Salmonella enterica* subsp. Typhimurium [58].

The functional profile obtained through Phylogenetic Investigation of Communities by Reconstruction of Unoserved States (PICRUSt) analyzed on MetaCyc showed some alterations on the inferred functional capabilities of the bacterial communities associated with phytobiotic-supplemented hens, in comparison with its unsuplemented counterpart (Figure 2). According to the observed in G2, phytobiotic supplementation could enhance some metabolic functions such as catechol degradation, creatinine degradation, L-leucine degradation, phospolipases activity, NAD biosynthesis, ectoine biosynthesis, formaldehyde assimilation by serine pathway, glycine betaine degradation, toluene degradation (via cathecol and *p*-cresol), starch degradation, L-arabinose degradation, and methanol oxidation to carbon dioxide. Some metabolic activities, such as fatty acid salvage and aromatic compound degradation, could be enhanced by phytobiotic supplementation when hens are challenged against *Salmonella*. Some specific pathways could be of interest in order to improve the productivity and health status of hens, such as creatinine metabolism observed in hens allocated in G2 group at week 30; this pathway could contribute to the storage of energy reserves in the muscles, which has a relationship to protein synthesis [59,60]. Ergothionein biosynthesis from L-histidine, observed in G4 group at week 40, could increase the antioxidant activity associated to ergothionein, which contributes to safeguard the cells from reactive oxygen species and pathogenic microorganisms [61,62]. On the other hand, degradation of starch pathway stimulation could contribute to obtaining glucose from the ration to be used as an energy source [63,64]. Meanwhile, some pathways such as amino acid degradation and biosynthesis of NAD+ could contribute to acetyl-Coa and energy generation that could help to the maintenance of several metabolic pathways [65,66,67,68]. All these functional inferences could explain the improvement and maintenance of hen’s productivity and egg quality observed when birds were supplemented with the *Allium*-based phytobiotic. 

## 5. Conclusions

*Allium*-based phytobiotic supplementation in laying hens showed be a beneficial alternative on productivity and egg quality, mainly associated with changes in fecal-associated bacterial communities as an indicator of the modulation of hens’ gut microbiota. Gut microbiota modulation could also modify their functional profile by the activation of pathways related to energy generation, complex nutrient metabolism, or compounds degradation, among others, which could drive the observed improvements. The use of *A**llium*-based phytobiotics could be a suitable alternative of natural origin to replace growth-promoting antibiotics, improve hens’ performance, and mitigate negative effects associated with *Salmonella* infection, exhibiting favorable effects after week 10 of supplementation in young layers.

## Figures and Tables

**Figure 1 microorganisms-10-00117-f001:**
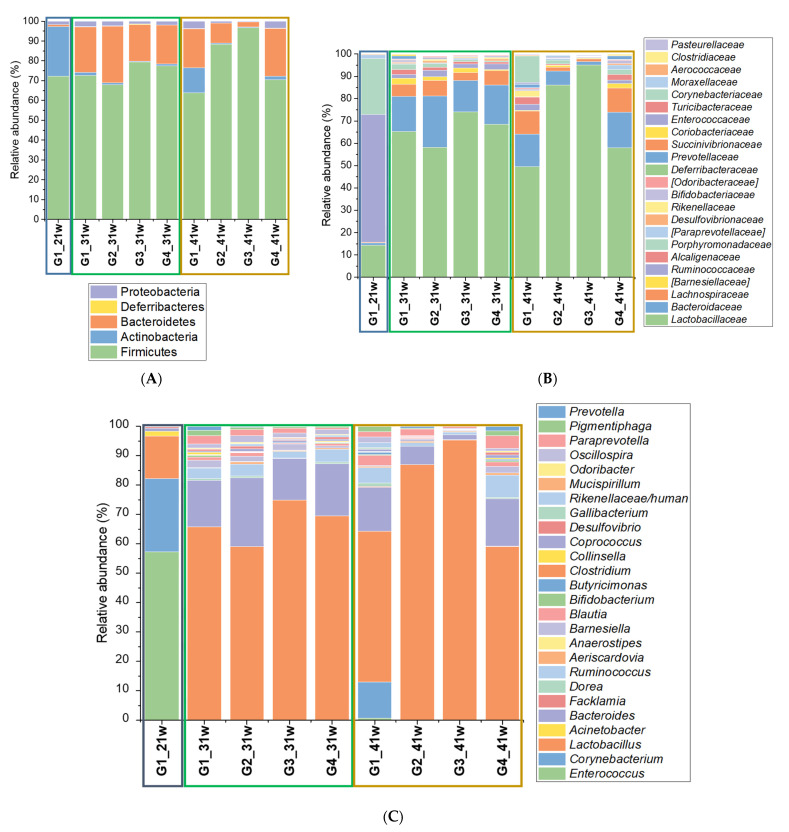
Bacterial communities associated with fecal samples obtained from laying hens supplemented and unsupplemented with an Allium-based phytobiotic, and challenged or unchallenged against *Salmonella* Pullorum. Bars represent the relative abundance at (**A**) *Phylum*, (**B**) Family, and (**C**) Genus level. G1: Control group; G2: *Salmonella* phytobiotic-supplemented unchallenged hens; G3: *Salmonella*-phytobiotic-supplemented challenged hens; G4: *Salmonella*-challenged hens. Samples were obtained from hens at 21, 31 and 41 weeks of age (21 w, 31 w and 41 w, respectively). The number of hens for each treatment was 48.

**Figure 2 microorganisms-10-00117-f002:**
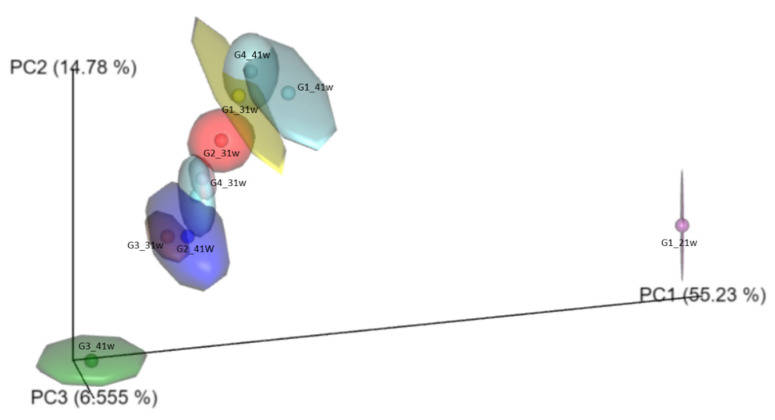
Beta diversity calculated for bacterial communities associated with fecal samples obtained from laying hens supplemented and unsuplemented with an *Allium*-based phytobiotic, and challenged or unchallenged against *Salmonella* Pullorum. G1: Control group; G2: *Salmonella*-phytobiotic-supplemented unchallenged hens; G3: *Salmonella*-phytobiotic-supplemented challenged hens; G4: *Salmonella*-challenged hens. Samples were obtained from hens at 21, 31, and 41 weeks of age (21 w, 31 w, and 41 w, respectively). The number of hens for each treatment was 48.

**Figure 3 microorganisms-10-00117-f003:**
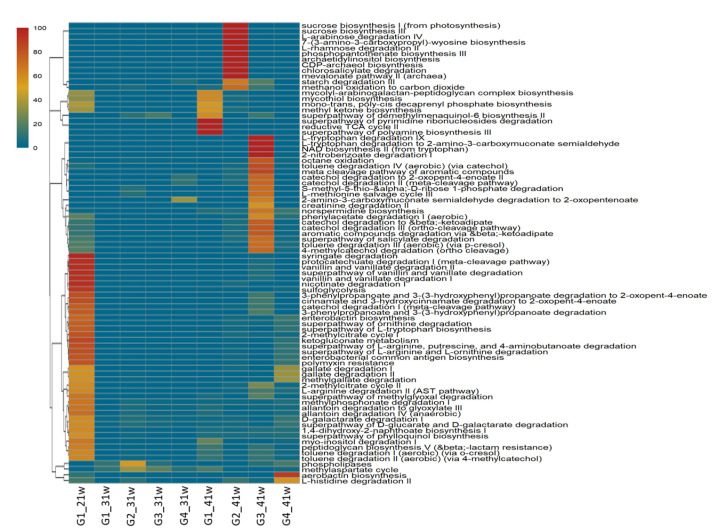
Heatmap representing the functional profiles of fecal-associated bacterial communities predicted by PICRUSt2 and MetaCyc. G1: Control group; G2: *Salmonella*-phytobiotic-supplemented unchallenged hens; G3: *Salmonella*-phytobiotic-supplemented challenged hens; G4: *Salmonella*-challenged hens. Samples were obtained from hens at 21, 31, and 41 weeks of age (21 w, 31 w, and 41 w, respectively). The number of hens for each treatment was 48.

**Figure 4 microorganisms-10-00117-f004:**
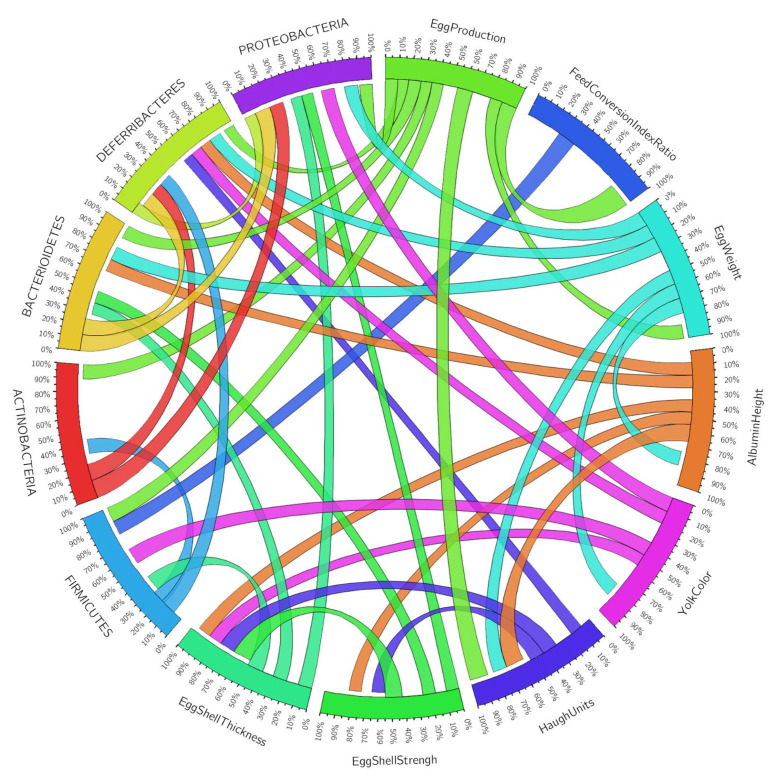
Schematic representation of the Pearson correlations (*p* > 0.05) between the relative abundance of the main *phyla* and the productive and egg quality variables. The number of hens for each treatment was 48 (Created with Circos Table Viewer v0.63–9).

**Table 1 microorganisms-10-00117-t001:** Productive parameters of *Salmonella* challenged and unchallenged laying hens with or without *Allium*-based phytobiotic supplementation.

Group	Age (Weeks)	Egg Production (Eggs/100 hens)	Feed Intake (g/day)	Feed Conversion Index Ratio	Mortality (%)
G1	21	76.23 ± 0.0223 ^a,B,A^	96.80 ± 0.8347 ^a,C^	2.67 ± 0.0621 ^a,A^	0 ± 0 ^a,A^
G2		72.91 ± 0.0220 ^a,B,A^	94.44 ± 0.0 ^a,C^	2.82 ± 0.1033 ^a,B,A^	0 ± 0 ^a,A^
G3		72.56 ± 0.0107 ^a,B^	85.58 ± 1.5274 ^a,B^	2.39 ± 0.0482 ^a,B,A^	0 ± 0 ^a,A^
G4		71.53 ± 0.0175 ^a,B^	91.48 ± 1.0434 ^a,C^	2.82 ± 0.0687 ^a,B^	0 ± 0 ^a,A^
G1	22–31	83.49 ± 0.002 ^a,B^	107.80 ± 0.218 ^a,A^	2.48 ± 0.009 ^b,A^	0 ± 0 ^a,A^
G2		81.51 ± 0.002 ^a,B^	105.77 ± 0.197 ^a,A^	2.59 ± 0.014 ^b,A^	0 ± 0 ^a,A^
G3		65.89 ± 0.002 ^b,B^	95.96 ± 0.176 ^b,A^	3.02 ± 0.019 ^b,a,A^	0.03 ± 0.002 ^a,A^
G4		63.93 ± 0.003 ^b,B^	98.48 ± 0.179 ^b,A^	3.661 ± 0.043 ^a,A^	0.03 ± 0.002 ^a,A^
G1	32–41	78.70 ± 0.001 ^b,A^	103.25 ± 0.096 ^a,B^	2.26 ± 0.005 ^a,B^	0 ± 0 ^a,A^
G2		79.32 ± 0.002 ^b,A^	98.165 ± 0.174 ^b,B^	2.28 ± 0.009 ^a,B^	0 ± 0 ^a,A^
G3		89.84 ± 0.001 ^a,A^	103.25 ± 0.128 ^a,A^	1.94 ± 0.003 ^b,B^	0 ± 0 ^a,A^
G4		82.82 ± 0.002 ^b,A^	106.79 ± 0.199 ^a,B^	2.232 ± 0.006 ^a,B^	0 ± 0 ^a,A^

^a,b,c^ Means in the same column, corresponding to the same birds age, that do not share a letter are significantly different (*p* < 0.05). G1: Control group; G2: Phytobiotic-supplemented *Salmonella*-unchallenged hens; G3: Phytobiotic-supplemented *Salmonella*-challenged hens; G4: *Salmonella*-challenged hens. ^A,B,C^ Means corresponding to the same group, but at different age, that do not share a letter are significantly different (*p* < 0.05). The number of hens for each treatment was 48.

**Table 2 microorganisms-10-00117-t002:** Egg quality of *Salmonella*-challenged and unchallenged laying hens with or without *Allium*-based phytobiotic supplementation.

	Age (w)	Egg Weight (g)	Albumin Height (mm)	Yolk Color (−)	Albumin Quality (Haugh units)	Egg Shell Strength (Kgf)	Egg Shell Thickness (mm)
G1	21	51.55 ± 0.271 ^a,B^	7.83 ± 0.066 ^a,A^	8.18 ± 0.048 ^a,B^	90.22 ± 0.295 ^a,A^	6.21 ± 0.052 ^a,A^	0.405 ± 0.002 ^a,A^
G2		49.17 ± 0.183 ^a,B^	6.04 ± 0.085 ^b,B^	7.83 ± 0.069 ^a,B^	80.55 ± 0.538 ^b,A^	5.87 ± 0.090 ^a,A^	0.396 ± 0.002 ^a,A^
G3		49.34 ± 0.206 ^a,C^	5.85 ± 0.080 ^b,B^	7.41 ± 0.097 ^a,B^	79.14 ± 0.616 ^b,B^	5.62 ± 0.075 ^a,A^	0.398 ± 0.002 ^a,A^
G4		49.98 ± 0.459 ^a,B^	6.39 ± 0.115 ^b,B^	8.16 ± 0.069 ^a,B^	82.14 ± 0.809 ^b,B^	5.47 ± 0.081 ^a,A^	0.398 ± 0.002 ^a,A^
G1	31	61.74 ± 0.343 ^b,a,A^	8.72 ± 0.108 ^a,A^	11.66 ± 0.041 ^a,A^	92.50 ± 0.591 ^a,A^	4.83 ± 0.055 ^a,B^	0.350 ± 0.002 ^a,B^
G2		59.18 ± 0.443 ^b,A^	8.27 ± 0.141 ^a,A^	8.83 ± 0.092 ^b,A^	90.40 ± 0.818 ^a,A^	4.54 ± 0.129 ^a,B,A^	0.344 ± 0.002 ^a,B^
G3		63.77 ± 0.221 ^a,A^	8.91 ± 0.126 ^a,A^	8.66 ± 0.108 ^b,A^	92.67 ± 0.803 ^a,A^	4.73 ± 0.104 ^a,B,A^	0.361 ± 0.003 ^a,B^
G4		61.94 ± 0.363 ^b,a,A^	9.02 ± 0.153 ^a,A^	11.66 ± 0.054 ^a,A^	93.04 ± 0.995 ^a,A^	4.87 ± 0.057 ^a,B,A^	0.358 ± 0.002 ^a,B^
G1	41	61.95 ± 0.437 ^a,A^	8.04 ± 0.211 ^a,A^	11.63 ± 0.158 ^a,A^	87.76 ± 1.218 ^a,A^	4.29 ± 0.108 ^a,B^	0.350 ± 0.002 ^a,B^
G2		58.96 ± 0.582 ^a,A^	6.76 ± 0.225 ^a,B,A^	7.45 ± 0.094 ^b,B^	80.73 ± 1.579 ^a,A^	4.23 ± 0.116 ^a,B^	0.356 ± 0.003 ^a,B^
G3		58.48 ± 0.308 ^a,B^	7.29 ± 0.123 ^a,B,A^	7.45 ± 0.137 ^b,B^	83.60 ± 1.416 ^a,B,A^	4.35 ± 0.085 ^a,B^	0.339 ± 0.002 ^a,B^
G4		60.28 ± 0.426 ^a,A^	8.11 ± 0.175 ^a,A^	11.90 ± 0.094 ^a,A^	89.69 ± 0.622 ^a,B,A^	4.26 ± 0.114 ^a,B^	0.355 ± 0.004 ^a,B^

^a,b,c^ Means in the same column, corresponding to the same age of birds that do not share a letter are significantly different (*p* < 0.05). G1: Control group; G2: *Salmonella*-unchallenged hens, phytobiotic supplemented; G3: *Salmonella*-challenged hens, phytobiotic supplemented; G4: *Salmonella*-challenged hens. ^A,B,C^ Means corresponding to the same group, but at different age, that do not share a letter are significantly different (*p* < 0.05). The number of hens for each treatment was 48.

**Table 3 microorganisms-10-00117-t003:** Number of sequences and diversity estimators for the different hen’s fecal samples.

Treatment	Age (w)	Raw Reads	Filtered Reads	Chao1	Shannon	Simpson
G1	21	348,114	78,428	11	1.24	0.6
G1	31	130,675	36,487	67	2.84	0.85
G2	31	415,451	106,401	106	3.25	0.87
G3	31	337,005	84,476	98	2.55	0.74
G4	31	202,520	60,949	92	2.91	0.85
G1	41	405,511	106,061	80	3.32	0.92
G2	41	224,088	53,304	61	2.07	0.67
G3	41	556,605	147,318	67	1.52	0.52
G4	41	299,560	74,802	74	3.27	0.91

## Data Availability

Not applicable.

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
