# Peer review of "Allium-Based Phytobiotic for Laying Hens’ Supplementation: Effects on Productivity, Egg Quality, and Fecal Microbiota"

_microorganisms, 2022, doi:10.3390/microorganisms10010117_

Round 1
Reviewer 1 Report
Major comments:
- How authors selected this dosage of Allium-based phytobiotic used in this work? And the major composition of Allium-based phytobiotic should be provided or showed (such as HPLC results).
- Besides 16s rRNA analysis of gut microbiota composition, SCFAs analysis should be carried out to study the effect of Allium-based phytobiotic on microbial metabolic product.
- If the results of functional prediction have some correlation with egg productivity. The correlation analysis was suggested.
- The Figure 1 showed that the results did not correspond with real results in the manuscript. For example, phytobiotic supplementation had no positive effect on the egg weight (Table 2), but the figure 1 showed a opposite result.
- The entire manuscript should be checked carefully for grammatical mistakes.
Minor comments:
- Line 51-55, This sentence should be added to the references.
- Line 59, The references are out of order, please check the entire manuscript. For example, [26].
- β-diversity was conducted by principal coordinate analysis (PCoA). Please supplement the method of PCoA.
Author Response
ATTACHED FILE WITH RESPONSES REVIEWER 1

Reviewer 2 Report
The current research study investigated potential use of an Allium-based phytobiotic product as a feed additive in laying hens. This study has indicated a positive effect of feeding this phytobiotic on productivity, which could be associated with gut microbiota modulation. The topic meets the Scope of Agricultural microbiology in Microorganisms. The aim of the study is clear and the experiment design is complete. Results were interpreted appropriately and comprehensively discussed. Specific comments for this manuscript are listed below.
Specific comments
- It would be better to add “fecal” before “microbiota” in the title to be more specific and to avoid confusion, giving this study evaluates the microbiota structure in feces rather than in intestinal segments.
- It is more accurate to use “egg shell strength” and “egg shell thickness” than “egg strength” and “egg thickness”, respectively, for example in Line 22 of Abstract. Please check other places and make the correction.
- In Line 72, does “gr” mean “g”? If it is, please revise “gr” to “g” because the latter is the SI unit symbol for “gram”.
- In the Section 3. Salmonella infection, the time point of Salmonella infection should be clarified.
- Does a manure sample is collected from each hen or a mixture of each cage (Line 110)? The source of each fecal sample for microbiota profiling should be specified.
- In Line 114, “DNA was stored at 20 °C until…”? Please check the sample storage temperature.
- The number of hens per treatment should be added in the legends of Tables and Figures.
- The text font size in Figure 3 is too small to read. It is recommended to make to font size larger to make visualization easier. Meanwhile, it would be better to add predicated pathway data in Figure 3 in the Supplemental Material.
- Delete “According to the results,” in Line 435.
Author Response
ATTACHED FILE WITH RESPONSES TO REVIEWER 2

Reviewer 3 Report
Minor comments:
- Bacteria phylum names should be in italics, shouldn't they? This question refers to the abstract and the text.
- Line 19: “Salmonella Pullorum” is it correct? Is it species or disease terms mixed?
- Please consider not using the term "positive effect" because it is not specific - what exactly does "positive" mean? Data is not feelings - it is data that changes in ways we can measure and describe. And data description should be simple, concise and specific. The same goes for "slightly" - slightly affected? What does that even mean? How much is slightly? Be more precise in describing the results.
- Line 21: “…the number of eggs laid and on the feed conversion ratio (p < 0.05), …” shouldn’t it be “…the number of eggs laid and on the feed conversion ratio (p<0.05, respectively), …”???
- Introduction - no clearly stated hypothesis within the study. What did the authors expect? We know the purpose, but why such a purpose?
- M&M section - 2.1. Phytobiotic mixture - It would be good to have at least the main ingredients listed. Again, "mostly" is not scientifically specific information.
- Line 77 – please provide the name of the institution where this bioethics and biosafety committee is located.
- Line 95 – intravenous administration? Is this correct? And if so, why this route?
- Line 114 – “20°C” – not “-20°C”???
- Lines 157-159 belongs to M&M section, not results.
- I suggest omitting superscript letters in Tables 1 and 2 where changes are not significant. This would make the results presented easier to perceive, making the differences presented more clearly visible.
- Lines 165-177 – Please do not repeat exact data from tables in the text - this is repetition of results and should be avoided. Please use % or proportions to describe such data in the text.
- Figure 3 has low resolution, text is not clear enough, difficult to read
Author Response
ATTACHED FILE WITH RESPONSES TO REVIEWER 3

Round 2
Reviewer 1 Report
This is an interesting study. I recommend acceptance of the manuscript.